# The Role of ACE2 in Neurological Disorders: From Underlying Mechanisms to the Neurological Impact of COVID-19

**DOI:** 10.3390/ijms25189960

**Published:** 2024-09-15

**Authors:** Jingwen Li, Xiangrui Kong, Tingting Liu, Meiyan Xian, Jianshe Wei

**Affiliations:** 1Institute for Brain Sciences Research, School of Life Sciences, Henan University, Kaifeng 475004, China; 2Institute for Sports and Brain Health, School of Physical Education, Henan University, Kaifeng 475004, China

**Keywords:** ACE2, neurological disorders, neuroprotection, neurodegenerative diseases, COVID-19

## Abstract

Angiotensin-converting enzyme 2 (ACE2) has become a hot topic in neuroscience research in recent years, especially in the context of the global COVID-19 pandemic, where its role in neurological diseases has received widespread attention. ACE2, as a multifunctional metalloprotease, not only plays a critical role in the cardiovascular system but also plays an important role in the protection, development, and inflammation regulation of the nervous system. The COVID-19 pandemic further highlights the importance of ACE2 in the nervous system. SARS-CoV-2 enters host cells by binding to ACE2, which may directly or indirectly affect the nervous system, leading to a range of neurological symptoms. This review aims to explore the function of ACE2 in the nervous system as well as its potential impact and therapeutic potential in various neurological diseases, providing a new perspective for the treatment of neurological disorders.

## 1. Introduction

More than 700 million people have been infected by the global COVID-19 pandemic, and over 6.8 million deaths from the disease have been reported (World Health Organization, March 2023 global pandemic situation), making it the most significant challenge in the history of human health [1]. COVID-19 patients most commonly experience symptoms such as fever, cough, and difficulty in breathing [2], but some exhibit initial neurological symptoms such as headache, cerebral hemorrhage, and infarction, among other neurological disorders [3]. These epidemiological findings are consistent with the ability of SARS-CoV-2 to penetrate the central nervous system by various means, leading to neural invasion and neuroticism. It can directly and/or indirectly infect neurons and glial cells and may trigger the development of neurological diseases (neurotoxicity) [4,5,6]. Angiotensin-converting enzyme 2 (ACE2) is a critical membrane-bound receptor and enzyme that serves dual functions in human physiology and viral infections. As a receptor, ACE2 is essential for the entry of several human coronaviruses. SARS-CoV-2 has been proven to share the same functional receptor with severe acute respiratory syndrome coronavirus (SARS-CoV), leading to the prevalence of COVID-19 and severe acute respiratory syndrome (SARS) in 2002–2003 [7,8]. After the outbreak of COVID-19, researchers found that ACE2 mediates the entry of SARS-CoV-2 into the human body [9]. SARS-CoV-2 relies on transmembrane serine protease 2 (TMPRSS2) and furin to lyse and activate SARS-CoV-2 envelope spike protein (S protein), which binds to its receptor ACE2 to infect host cells [10,11,12]. As an enzyme, ACE2 is a homologous molecule of angiotensin-converting enzyme (ACE) that functions as an enzyme (monocarboxypeptidase) [13,14]. In addition to acting as a receptor for SARS-CoV and SARS-CoV-2, ACE2 also plays a crucial role in the renin–angiotensin system (RAS) [15], hydrolyzing angiotensin II (Ang II) into angiotensin (1–7) (Ang 1-7) [16], which binds to the Mas receptor to exert vasodilatory, anti-inflammatory, and antifibrotic effects [17]. Multiple studies have shown that a SARS-CoV infection can affect the expression of ACE2 on cells, disrupt the physiological balance between ACE/ACE2 and Ang II/Ang 1-7, and cause serious organ damage [18,19,20].

The expression of ACE2 in healthy human tissues has been described by databases such as Protein Atlas, with the highest expression detected in the intestine, testes, lungs, cornea, heart, kidneys, and adipose tissue [21,22], and it is also present in the brain [23]. It participates in the neural regulation of normal physiological functions, such as metabolic activity, stress response, and neurogenesis [24,25]. ACE2 is expressed in neurons and glial cells of the brain, especially in the brainstem and cardiovascular regulatory regions, including the solitary tract nucleus, paraventricular nucleus, and ventrolateral medullary nucleus [1]. However, the expression of ACE2 is relatively high in certain neurovascular unit components, especially in peri-brain cells. These cells originate from neural crest stem cells, and they physically connect endothelial cells and astrocytes, thereby promoting their own maturation and the production of basement membrane components [26,27]. The interaction between ACE2 and viral proteins in the cerebral vascular endothelium may disrupt the blood–brain–barrier (BBB), causing endothelial damage, facilitating viral entry into the brain, and leading to adverse effects on the central nervous system [28,29,30,31]. In light of the above, an exploration and summary of the exact role played by ACE2 in neurological diseases may serve as a valuable guide for further research.

## 2. Biological Characteristics of ACE2 and Its Association with the Nervous System

In 2000, while investigating new genes associated with heart failure, two research teams independently used different methods to clone the same gene from heart samples of patients with idiopathic dilated cardiomyopathy and human lymphoma samples. They named the gene ACE2 and ACEH, respectively, later collectively referred to as ACE2 [32]. ACE2, identified as the first reported homologue of ACE, began to attract attention, and subsequent studies were conducted [33,34,35]. Early research indicated that ACE2 was highly expressed in the heart, kidneys, and testes, with low expression in other tissues [36]. However, further research revealed that ACE2 is also expressed in the jejunum, duodenum, cecum, ileum, and various other tissues, including the lungs, bone marrow, spleen, liver, ovaries, brain, adipose tissue, and macrophages [37,38,39].

### 2.1. Enzymatic Activity of ACE2

ACE2 is a multifunctional metalloprotease belonging to the M2 family. It operates through a zinc-dependent catalytic mechanism, playing a role in various biological processes [40]. ACE2 possesses a broad capacity to hydrolyze cardiovascular-active peptides [7]. Studies have shown that ACE2 can hydrolyze angiotensin I (Ang I) into angiotensin (1–9) (Ang 1-9) and more efficiently hydrolyze Ang II into Ang 1-7 [36]. Additionally, ACE2 exhibits high catalytic activity towards other vasoactive peptides such as apelin [41], which contracts blood vessels and protects cardiovascular function, highlighting ACE2’s significant role in cardiovascular regulation [42,43]. ACE2 also participates in degrading des-Arg9-bradykinin, which binds to B1 receptors expressed in damaged and inflamed tissues, promoting local vasodilation [44,45]. Therefore, ACE2 is a versatile enzyme critical for inflammation and cardiovascular function.

### 2.2. ACE2 in the Regulation of the RAS System

The RAS is expressed in various tissues throughout the body [46]. Renin converts angiotensinogen, produced by hepatocytes, into the inactive peptide Angiotensin I (Ang I). Angiotensin-converting enzyme (ACE) then cleaves Ang I into the active peptide Angiotensin II (Ang II), which signals through the type 1 angiotensin receptor (AT1R). This signaling induces oxidative stress, promotes inflammation, and causes vasoconstriction, leading to elevated blood pressure. This pathway is known as the classical RAS pathway, or the ACE/Ang II/AT1R axis [47,48]. ACE2, a key regulator of the RAS, enzymatically converts Ang II into Ang 1-7, which exerts vasodilatory effects by binding to the Mas receptor (MasR), forming the ACE2/Ang 1-7/MasR axis [49,50]. This axis not only mitigates the vasoconstrictive effects of Ang II but also activates MasR signaling pathways that promote vasodilation and reduce blood pressure [51]. The counter-regulatory axis of the RAS and the ACE2/Ang 1-7/MasR axis, counteracts the effects of the ACE/Ang II/AT1R axis and has been shown to aid in the repair of organ damage in cardiovascular and renal diseases [52]. Thus, ACE2 plays a crucial role in maintaining cardiovascular homeostasis and offers new therapeutic perspectives and potential targets for treating hypertension and related cardiac and renal diseases. The specific regulatory mechanism is shown depicted in Figure 1A.

### 2.3. ACE2 as the Receptor for SARS-CoV and SARS-CoV-2

ACE2 plays a critical role as a receptor in the infection process of coronaviruses, including SARS-CoV and SARS-CoV-2. It is a membrane-bound protein that is expressed on the surface of cells in various tissues, such as the central nervous system. The S protein on the surface of these viruses binds to ACE2 receptors on host cells, initiating the fusion of viral and host cell membranes, which allows the virus to enter the host cell for replication and spread [11,53]. Consequently, ACE2 is a significant target for developing strategies to prevent and treat coronavirus infections [54,55].

## 3. ACE2 and the Nervous System

Recent research has revealed that ACE2 plays a significant role in the nervous system, impacting neuroprotection, neurodevelopment, and neuroinflammation.

### 3.1. Brain Expression

ACE2 expression in the brain is predominantly localized in specific regions, including the pons, visual cortex, amygdala, choroid plexus of the thalamus, and paraventricular nucleus [23,56]. In the brains of Alzheimer’s disease (AD) patients, ACE2 protein levels are downregulated in the basal ganglia, hippocampus, entorhinal cortex, and middle frontal gyrus [57]. Additionally, ACE2 expression is not restricted to a single type of neural cell but is found across various types, such as cortical neurons, astrocytes, oligodendrocytes, microglia, and endothelial cells [7,58,59,60]. In neurons, ACE2 may regulate neurotransmitter release and synaptic plasticity, thereby influencing learning and memory processes [61]. In astrocytes, it might be involved in modulating neuroinflammation and maintaining extracellular matrix homeostasis [62]. In oligodendrocytes, ACE2 could affect myelination and nerve conduction velocity [63]. In cerebrovascular endothelial cells, it may contribute to the integrity of the BBB [64]. These findings suggest that ACE2 may play a role in the development and progression of neurological diseases and symptoms.

### 3.2. ACE2 and Neuroprotection

The distribution of ACE2 in the brain plays a crucial role in protecting the nervous system and maintaining neural function, primarily through its neuroprotective effects [65,66]. ACE2 converts Ang II to Ang 1-7, which has vasodilatory and anti-inflammatory properties, thereby reducing oxidative stress and inflammation in neural cells [60,67]. Additionally, ACE2 expression in cerebrovascular endothelial cells helps maintain the integrity of the BBB, preventing toxins and pathogens from reaching the nervous system [63,68]. Its expression in key brain regions, such as the hippocampus and cerebral cortex, may regulate cognitive function, emotional states, and learning and memory processes [66,67,68,69,70,71]. Moreover, ACE2 may promote neurogenesis in the dentate gyrus of the hippocampus, which is significant for brain plasticity and repair [1].

## 4. ACE2 and Neurological Disorders

### 4.1. ACE2 and AD

AD is a neurodegenerative disorder characterized by insidious onset, progressive cognitive impairment, decline in daily social functioning, and neuropsychiatric symptoms. It is the most common cause of dementia. The primary pathological features of AD include the deposition of amyloid-β (Aβ) protein and hyperphosphorylation of tau protein, leading to neurofibrillary tangles. These misfolded proteins further impair synaptic function, disrupt brain network connectivity, and result in cognitive deficits [72,73]. AD is closely associated with aging, and ACE2 expression is downregulated in the brain microvasculature and brain tissue of aged mice [74]. Studies have shown that ACE2 levels are reduced in the basal nucleus, hippocampus, entorhinal cortex, medial frontal gyrus, visual cortex, and amygdala of AD patients. This reduction correlates negatively with Aβ levels and phosphorylated tau (p-tau) pathology. Furthermore, individuals carrying the APOEε4 allele, which is associated with AD, also exhibit significantly lower ACE2 levels, suggesting that ACE2 may play a role in AD-related mechanisms [74,75,76]. The ACE2 activator dimethylamine acetylurea (DIZE) significantly reduces Aβ1-42 levels, hyperphosphorylated tau, and pro-inflammatory cytokines in the brains of senescence-accelerated mouse-prone 8 (SAMP8) mice, alleviating synaptic and neuronal loss and improving cognitive function [77]. DIZE also significantly lowers Aβ1-42 levels in the hippocampus of APP^Swe^ mutant Tg2576 mice and restores cognitive function by modulating MasR and NMDA NR2B receptors and their downstream signaling pathways [69]. Additionally, DIZE upregulates miR-224-5p in astrocytes, significantly reducing cognitive impairment and neural damage in APP/PS1 mice while inhibiting pro-inflammatory cytokines and the NLRP3 inflammasome [78]. ACE2 is also found to inhibit neurotoxicity by converting Aβ43 (an early and highly amyloidogenic form) to Aβ42, which is further cleaved into less toxic Aβ40 and Aβ41 [75]. Moreover, ACE2 improves pathological angiogenesis and BBB damage in AD models by inhibiting the NF-κB/VEGF/VEGFR2 pathway, making it a potential therapeutic target for endothelial dysfunction in AD [79].

### 4.2. ACE2 and Parkinson’s Disease

Parkinson’s disease (PD) is the second most common rapidly progressive neurodegenerative disorder, characterized clinically by both motor and non-motor symptoms. The primary pathological features of PD include the loss of dopaminergic neurons in the substantia nigra and the accumulation of misfolded α-synuclein protein [80,81]. Studies have shown significantly elevated levels of ACE2 autoantibodies (ACE2-AA) in PD patients, which are correlated with various inflammatory factors and metabolites, suggesting a potential role of ACE2 in PD [82]. Research using C57BL/6J wild-type and ACE2 knockout mouse models revealed that a ACE2 deficiency exacerbates MPTP-induced motor and emotional deficits in PD model mice, as well as inflammation and oxidative stress [83]. Moreover, ACE2 activation in LPS-induced BV2 cells demonstrated anti-inflammatory and neuroprotective effects by inhibiting microglial activation through the suppression of inflammation and reactive oxygen species production [66]. It has been suggested that ACE2 not only plays a critical role in the RAS but may also regulate oxidative stress and inflammatory responses in dopaminergic neurons via the mitochondrial ACE2/MrgE/NO axis, which could significantly impact neurodegenerative processes [60]. Furthermore, DIZE treatment improved motor function in 6-OHDA-induced PD rat models, provided neuroprotection to dopaminergic neurons, and reduced neuroglial activation and neuroinflammation. These protective effects are associated with DIZE-induced ACE2 activation, resulting in the production of Ang 1-7 and subsequent MasR receptor-mediated actions [84].

### 4.3. ACE2 and Ischemic Stroke

Ischemic stroke (IS) is a common cerebrovascular disease and the second leading cause of death worldwide [85]. The role of ACE2 in IS involves multiple mechanisms, including direct neuroprotection and the regulation of inflammatory responses. ACE2 enzymatically converts Ang II to the protective peptide Ang 1-7, which helps reduce Ang II-induced vasoconstriction and inflammation, thereby lowering the risk of IS [86]. Ang 1-7, known for its neuroprotective properties, mitigates oxidative stress, promotes cell survival, and enhances neuronal survival through the activation of survival signaling pathways such as PI3K/Akt and ERK/MAPK [87]. Studies have shown that upregulating ACE2 levels can activate endothelial nitric oxide synthase (eNOS) and neuronal nitric oxide synthase (nNOS), increasing nitric oxide (NO) production in endothelial cells and neurons. This reduces oxidative stress and maintains neurovascular homeostasis in IS [74]. Additionally, neuronal overexpression of ACE2 can decrease reactive oxygen species (ROS) production by downregulating Nox2/Nox4 expression, thereby reducing oxidative stress and improving neurological function in IS [88]. ACE2 also lowers the release of inflammatory cytokines, reducing further brain tissue damage [89]. M1-like microglia exhibit cytotoxicity, and their activation induces the production and release of pro-inflammatory cytokines such as tumor necrosis factor-α (TNF-α), matrix metalloproteinases (MMPs), interleukin-1β (IL-1β), and interleukin-6 (IL-6), exacerbating tissue inflammation during the acute phase of IS [90]. By hydrolyzing Ang II to Ang 1-7, ACE2 can inhibit the inflammation process triggered by the activation of M1-like microglia during IS, thereby protecting brain tissue [91].

### 4.4. ACE2 and Depression, Anxiety

Studies have shown that overexpression of ACE2 in the basolateral amygdala reduces anxiety-like behavior by activating MasR, which affects GABAergic neurotransmission [92]. Additionally, the overexpression of ACE2 in corticotropin-releasing hormone (CRH) cells can attenuate the activation of the hypothalamic–pituitary–adrenal (HPA) axis, thereby alleviating anxiety-like behavior [93]. In male mice with ACE2 overexpression, increased ACE2 can inhibit stress-induced activation of the HPA axis via CRH expression. However, in female mice, increased ACE2 primarily exerts an anxiolytic effect without reversing HPA axis activity [94]. The specific regulatory mechanism is shown depicted in Figure 1B.

## 5. COVID-19 and ACE2 in Neurological Diseases

The COVID-19 pandemic, caused by SARS-CoV-2, has become a global health crisis [95]. While primarily affecting the respiratory system, COVID-19 also impacts multiple organ systems, notably the nervous system [96]. Increasing evidence links SARS-CoV-2 infection to various neurological disorders [97,98]. Reports indicate that COVID-19 patients exhibit numerous neurological symptoms, including headaches, dizziness, altered consciousness, strokes, encephalitis, neuromuscular diseases, and a loss of smell and taste [99,100]. ACE2, the key receptor for SARS-CoV-2 entry into host cells, has garnered attention for its role in neurological diseases. The manifestation of these neurological symptoms suggests that SARS-CoV-2 may directly or indirectly affect the nervous system via the ACE2 receptor [7].

SARS-CoV-2 uses its S protein to bind to the N-terminal of the ACE2 receptor on host cells. This binding, facilitated by host cell proteases such as TMPRSS2, induces conformational changes in the S protein, promoting fusion with the host cell membrane and subsequent viral entry into the cell [54]. Once inside, the virus can enter neurons or glial cells, increasing intracellular viral load and triggering cellular responses such as apoptosis and autophagy, exacerbating neuronal damage [99,101].

ACE2 also plays a crucial role in regulating the immune response in the nervous system. Under normal physiological conditions, ACE2 modulates the RAS balance through its enzymatic activity, impacting neuroinflammatory responses. However, during SARS-CoV-2 infection, the binding of the virus to ACE2 inhibits its enzymatic activity, disrupting RAS balance. This disruption leads to elevated levels of pro-inflammatory Ang II and reduced levels of anti-inflammatory Ang 1-7, aggravating neuroinflammation [102]. Consequently, more immune cells, including microglia and astrocytes, infiltrate the infection site, releasing inflammatory cytokines and chemokines, which further damage neurons [103]. Furthermore, SARS-CoV-2 infection may cause the shedding of ACE2 into its soluble form (sACE2) via the hydrolytic action of proteases such as TMPRSS2. This process could have significant implications for viral transmission and host signaling [104]. While the exact role of sACE2 remains under investigation, it is likely crucial in regulating the RAS and influencing COVID-19 severity. Therefore, both the membrane-bound and soluble forms of ACE2 should be comprehensively considered when evaluating the impact of SARS-CoV-2 infection on the nervous system.

ACE2 is also critical for maintaining the integrity and function of the BBB. The BBB protects the brain from pathogens and harmful substances, and ACE2 expression in BBB endothelial cells is vital for maintaining tight junctions and permeability [101]. When investigating the role of ACE2 in SARS-CoV-2 infection, it is essential to consider not only its function as a viral receptor but also the regulation of its expression and downstream effects in pathological conditions [105]. ACE2 activity is influenced by its expression levels and post-translational modifications, including glycosylation, phosphorylation, and ubiquitination. These modifications can affect ACE2 stability, subcellular localization, and binding affinity to ligands or interacting proteins, indirectly impacting the balance between Ang II and Ang 1-7. SARS-CoV-2 infection may further exacerbate the imbalance between Ang II and Ang 1-7 by disrupting the post-translational modification process of ACE2 [106]. Additionally, sACE2, an enzymatic product of ACE2, plays a crucial role in regulating the Ang II/Ang 1-7 balance. sACE2 can diffuse freely in circulation and competitively bind Ang II with membrane-bound ACE2, modulating the activity of Ang II and its downstream effects [107]. SARS-CoV-2 infection may also alter the balance between Ang II and Ang 1-7 by affecting sACE2 production or function, which in turn influences the permeability of the BBB. The increased permeability of the BBB may involve the coordinated action of several mechanisms, including ACE2 downregulation, elevated Ang II levels, the release of inflammatory mediators, and disruption of tight junctions [68,108,109]. Thus, understanding the impact of SARS-CoV-2 infection on BBB permeability requires a comprehensive analysis of ACE2 expression regulation, post-translational modifications, sACE2 activity, and inflammatory mediator interactions. These factors collectively contribute to BBB dysfunction and the neurological complications associated with infection.

In summary, ACE2 plays a multifaceted role in the pathogenesis of SARS-CoV-2-related neurological diseases. It serves as the viral entry point and participates in modulating immune responses and maintaining BBB integrity. The specific regulatory mechanism is shown depicted in Figure 1C.

## 6. Discussion

In recent years, the role of ACE2 in the nervous system has garnered significant attention, particularly in the context of the COVID-19 pandemic. Our review explores the multifaceted functions of ACE2 in the brain, including its involvement in normal physiological processes, its potential as a therapeutic target, and its complex relationship with neurological diseases. Here, we delve into the implications of our findings and their impact on future research.

### 6.1. ACE2 as a Key Regulatory Factor in the Nervous System

ACE2 plays a pivotal role in the RAS and is of great significance in maintaining homeostasis within the nervous system. By catalyzing the conversion of Ang II to Ang 1-7, ACE2 balances the vasoconstrictive, pro-inflammatory, and profibrotic effects of Ang II, promoting vasodilation, anti-inflammation, and antifibrosis. This balance is crucial for brain function. Our review highlights the expression of ACE2 in the brain, including in neurons and glial cells, indicating its essential role in neural regulation. However, the COVID-19 pandemic uncovered the dark side of ACE2’s role in the nervous system, as SARS-CoV-2 uses ACE2 as its primary entry receptor, disrupting normal neural functions and leading to neurological symptoms.

### 6.2. ACE2 and SARS-CoV-2: A Double-Edged Sword

The discovery that SARS-CoV-2 uses ACE2 to enter host cells has greatly altered our understanding of ACE2’s role in the nervous system. On one hand, the fundamental function of ACE2 in maintaining RAS homeostasis is disrupted by viral infection, leading to inflammation, oxidative stress, and neurotoxicity. On the other hand, the expression levels and activity of ACE2 may be affected by SARS-CoV-2 infection, further complicating the pathological situation. The interaction between SARS-CoV-2 and ACE2 in the brain disrupts the BBB, facilitating viral entry into the central nervous system. This process triggers a series of inflammatory responses, resulting in neuronal damage and dysfunction, manifesting as a range of neurological symptoms observed in COVID-19 patients. Thus, in the context of SARS-CoV-2 infection, ACE2 is a double-edged sword. While its normal function is vital for neural health, the virus’s exploitation of it exacerbates neuropathology.

### 6.3. ACE2 and Neurological Symptoms of Long COVID

Multiple studies have reported the persistence of long-term symptoms following COVID-19 infection in some patients, including fatigue, dyspnea, headaches, musculoskeletal pain, altered taste and smell, brain fog, and cognitive decline (such as memory and reasoning difficulties), collectively known as “long COVID” [110,111,112,113,114]. The World Health Organization defines long COVID as symptoms that persist or emerge within three months of the initial SARS-CoV-2 infection, last for at least two months, and cannot be explained by any other cause [115]. Although the precise mechanisms underlying long COVID are not yet fully understood, existing research suggests that the lingering effects of ACE2 following SARS-CoV-2 infection may be closely associated with the neurological symptoms of long COVID [116]. Further studies are needed to explore the long-term alterations in ACE2 following viral infection and how these changes affect the homeostasis and repair processes of the nervous system. Specifically, ACE2’s vital roles in maintaining neurological homeostasis, such as regulating inflammatory responses, mitigating oxidative stress, and balancing neurotransmitter levels, may significantly contribute to the pathophysiological processes of long COVID. Additionally, therapies targeting ACE2 could offer promising treatment strategies for patients suffering from long COVID [117].

### 6.4. Therapeutic Potential of ACE2 in Neurological Diseases

The multifaceted roles of ACE2 in the nervous system suggest that it could serve as a therapeutic target for a variety of neurological diseases. The modulation of ACE2 activity could offer a novel treatment approach, particularly considering its neuroprotective and anti-inflammatory properties.

#### 6.4.1. Modulation of ACE2 Activity

The pharmacological modulation of ACE2 activity presents a promising avenue for therapeutic intervention. Small molecules or biologics that enhance the expression or activity of ACE2 could counteract the detrimental effects of Ang II, promoting neuroprotection. For instance, the ACE2 activator DIZE has been shown to reduce pathological progression in models of Alzheimer’s disease [67,75]. Gene therapy techniques, such as viral vector-mediated gene transfer, could also be employed to introduce the ACE2 gene into damaged neurons or glial cells, thereby increasing its expression levels. Studies in animal models using overexpression or knockout of the ACE2 gene have confirmed its significant role in the nervous system [81,86]. Additionally, small-molecule drugs are being explored to increase the expression or activity of ACE2; these compounds may achieve this by affecting the transcription, translation, or stability of ACE2 [100].

ACE2 generates Ang 1-7 by hydrolyzing Ang II, which then activates the Mas receptor and exerts anti-inflammatory effects. Therefore, interventions targeting the ACE2/Ang 1-7/Mas axis may hold potential for anti-inflammatory therapies. The development of drugs that can stabilize Ang 1-7 or activate the Mas receptor could become a novel approach for treating neuroinflammatory diseases. Moreover, by inhibiting the activity of ACE, the production of Ang II can be reduced, indirectly enhancing the anti-inflammatory effects of ACE2.

#### 6.4.2. Neurorestorative Therapies

The capacity of ACE2 to promote neurogenesis and repair within the brain suggests its potential in restorative therapies for neurodegenerative diseases. Strategies that harness the neurorestorative potential of ACE2 could aid in recovery following neural injury or in slowing the progression of degenerative diseases. ACE2 plays a critical role in maintaining the integrity of the BBB, and protecting or repairing an impaired BBB could become an important avenue for treating neurological diseases. The development of drugs that stabilize ACE2 expression in cerebrovascular endothelial cells, as well as the promotion of ACE2 upregulation through neural stem cell transplantation, may offer new strategies for the treatment of neurological diseases.

### 6.5. Future Research Directions

#### 6.5.1. Regulation of ACE2 Expression

Research should focus on the specific expression patterns and regulatory mechanisms of ACE2 in different types of cells (such as neurons and glial cells) and various brain regions. Understanding the upregulation or downregulation mechanisms of ACE2 expression through transcription factors and epigenetic modifications will aid in developing new strategies for the targeted regulation of ACE2 expression. Exploring the impact of environmental factors (such as inflammation, hypoxia, and oxidative stress) on ACE2 expression and how these influences act through neuro-immune-endocrine networks in the occurrence and progression of neurological diseases is essential.

#### 6.5.2. In-Depth Study of ACE2 and COVID-19-Related Neurological Symptoms

We recommend future studies to further refine the current understanding of the specific pathways and molecular mechanisms by which SARS-CoV-2 enters the nervous system through ACE2, especially how the virus affects key neuronal functions such as metabolism, signal transduction, and synaptic plasticity. Further research should conduct large-scale clinical trials to assess the efficacy of ACE2 inhibitors or activators in the treatment of neurological symptoms in COVID-19 patients and explore their optimal dosage for use. While the expression profile of ACE2 reveals its distribution across various tissues, the functional role of ACE2 extends beyond tissue-specific expression. Notably, sACE2 can be released into circulation, enabling SARS-CoV-2 infection even in tissues with limited ACE2 expression [118]. Studies have demonstrated that sACE2 can facilitate viral entry into cells lacking cellular ACE2, which has important implications for viral transmission and infection mechanisms [119]. This finding underscores the complexity of ACE2’s functional roles and emphasizes the need for further investigation into the mechanisms that enable its distant functionality. Consequently, while the tissue-specific expression of ACE2 is significant, understanding the role of its soluble form in disease pathogenesis is equally essential.

#### 6.5.3. Potential Therapeutic Role of ACE2 in Neurodegenerative Diseases

Considering the roles of ACE2 in neuroprotection, anti-inflammation, and antioxidation, future studies should investigate its potential therapeutic value in neurodegenerative diseases such as AD and PD. Through animal models and cellular experiments, we recommend researchers verify the preventive and therapeutic effects of ACE2 and its regulators on these diseases. We also recommend researchers explore the synergistic effects of ACE2 with other neuroprotective factors, such as brain-derived neurotrophic factor (BDNF) and glial cell-derived neurotrophic factor (GDNF), to develop more effective combination therapy plans.

#### 6.5.4. The Role of ACE2 in Neurodevelopment and Regeneration

Future researchers should study the specific roles of ACE2 in the neurodevelopmental process, including key steps such as neuronal migration, differentiation, and synapse formation. Future researchers should utilize gene-editing technologies (such as CRISPR/Cas9) to create animal models with ACE2 gene knockout or overexpression to reveal its specific mechanisms in neurodevelopment. Investigate the potential of ACE2 in neural regeneration, especially its application in neural injury repair and functional recovery. Future researchers should evaluate the impact of ACE2 and its regulators on the proliferation, differentiation, and migration of neural stem cells through in vitro cell culture and in vivo animal experiments.

#### 6.5.5. The Relationship between ACE2 and Cerebrovascular Diseases

We recommend future studies investigate the protective role and pathological mechanisms of ACE2 in cerebrovascular diseases (such as stroke and cerebral hemorrhage). Further research should analyze how ACE2 can mitigate the occurrence and development of cerebrovascular diseases by regulating blood pressure and improving vascular endothelial function. Future research should explore the potential application value of ACE2 in the treatment of cerebrovascular diseases, such as the development of new drugs or treatment methods based on ACE2, to improve patient treatment outcomes and quality of life.

In summary, the multiple roles of ACE2 in the nervous system and its potential impact on neurological diseases such as COVID-19 provide us with a new perspective for the treatment of neurological disorders. ACE2 not only acts as the receptor for SARS-CoV-2 and plays a key role in the process of viral invasion but also regulates the normal physiological functions of the nervous system through its enzymatic action, including metabolic activity, stress response, and neurogenesis. Therefore, a deep understanding of the molecular mechanisms and regulatory networks of ACE2 in the nervous system is of great significance for the development of new therapeutic strategies for neurological diseases.

Furthermore, considering the central role of ACE2 in regulating the balance of the renin-angiotensin system, modulating the expression or activity of ACE2 may help restore homeostasis in the nervous system and reduce neuroinflammation and neuronal damage. Future research should further explore the specific mechanisms of action of ACE2 in neurodegenerative diseases, cerebrovascular diseases, and neuropsychiatric diseases and assess the potential application value of ACE2 modulators in the treatment of these diseases. This will not only help deepen the understanding of the pathogenesis of neurological diseases but also provide a scientific basis for the development of more precise and effective treatment methods.

## 7. Conclusions

ACE2, as a multifunctional metallopeptidase, plays an important role in the neuroprotection, development, and regulation of inflammation. The COVID-19 pandemic has further highlighted the key position of ACE2 in the nervous system. Future research should delve into the regulatory mechanisms of ACE2 expression, its association with neurological symptoms related to COVID-19, its potential therapeutic role in neurodegenerative diseases, its role in neurodevelopment and regeneration, and its relationship with cerebrovascular diseases. These studies will not only help to reveal the complex functions and mechanisms of ACE2 in the nervous system but may also provide new ideas and methods for the prevention and treatment of neurological diseases. With the continuous advancement of science and technology and in-depth research, it is believed that research on ACE2 in the field of the nervous system will achieve more groundbreaking results and make a greater contribution to the cause of human health.

## Figures and Tables

**Figure 1 ijms-25-09960-f001:**
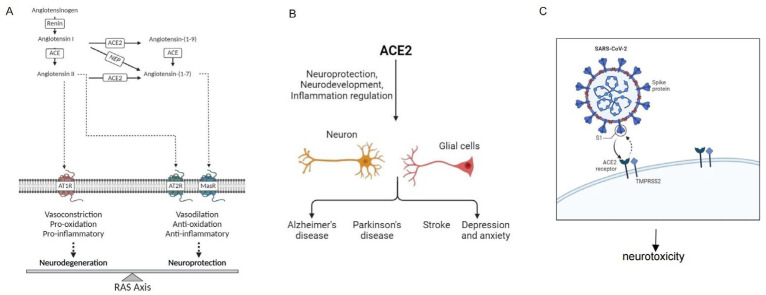
The role of ACE2 in neurological diseases and its impact on the nervous system under SARS-CoV-2 infection. (Created with Biorender.com). (**A**). Angiotensinogen is cleaved by renin to form Ang I. ACE converts Ang I to Ang II, which is the main effector peptide of RAS. Ang II triggers its cellular effects by activating its main receptors, AT1R and angiotensin II receptor 2 (AT2R), thereby counteracting the effects of AT1R activation. ACE2 cleaves Ang II to form Ang 1-7, activating MasR and counteracting Ang II-mediated effects. ACE2 also cleaves Ang I to form Ang 1-9, which is then cleaved by ACE to produce Ang 1-7. Ang 1-9 can also be formed and activated by neuropeptidases, such as NEP, to form AT2R. Some of the effects mediated by Ang 1-7 may also involve AT2R. (**B**). ACE2 not only directly participates in the protection of neurons, but also maintains the homeostasis of the nervous system by regulating inflammatory responses. It can promote anti-inflammatory reactions, inhibit the production of neurotoxic substances, participate in physiological processes such as neuronal vasodilation and antioxidant stress, and thus play an important role in neurological diseases such as AD, PD, IS, depression, and anxiety. (**C**). The SARS-CoV-2 virus enters host cells by binding to ACE2 receptors, exacerbating neuronal damage.

## Data Availability

Not applicable.

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
