# Peer review of "The Role of ACE2 in Neurological Disorders: From Underlying Mechanisms to the Neurological Impact of COVID-19"

_ijms, 2024, doi:10.3390/ijms25189960_

Round 1

Reviewer 1 Report

Comments and Suggestions for Authors

The article focuses on discussing the roles of ACE2 in SARS-CoV-2-induced neurological disorders. The topic is interesting and could be relevant to the development of long COVID. The authors should consider discussing long COVID in the writing.

ACE2 can be proteolytically by proteases, including ADAM17 and TMPRSS2, resulting in shedding the soluble form of ACE2. This form of ACE2 is important for the host to disseminate long-range signaling. Studies have shown that the downregulation of ACE2 expression could be due to the shedding of ACE2 by the activities of these proteases. Furthermore, clinical studies have detected a positive correlation between the detection of the soluble form of ACE2 and the severity of COVID-19. Therefore, the assumption of SARS-CoV-2 infection causes the downmodulation of the RAS signaling pathway could be inaccurate, as the soluble form of ACE2 is functionally active.

The authors cited Ref 21 to support the claim that the ACE2 expression could be detected in almost all human organs to various degree is inaccurate. Multiple studies have found that the expression of ACE2 is very restricted. It is recommended that the authors can reference data from general public databases, including https://www.proteinatlas.org/ and https://www.ncbi.nlm.nih.gov/geo/. However, the expression profile may not fully explain the functional sites of ACE2, as the soluble form of ACE2 could be functioning remotely in tissues that originally do not express ACE2. Indeed, some studies have shown that the soluble form of ACE2 could mediate SARS-CoV-2 infection in cells that do not express cellular ACE2. The authors should also discuss this in the manuscript.

Discussion 6.1. mentioned that “By catalyzing the conversion of Ang II to Ang 1-7, ACE2 balances the vasoconstrictive, pro-inflammatory, and profibrotic effects of Ang II, promoting vasodilation, anti-inflammation, and antifibrosis.”  The statement is true about the function of Ang1-7 which is involved in vasodilation, anti-inflammation, and antifibrosis. However, SARS-CoV-2 infection decreases the expression level of ACE2, which should reduce the level of Ang1-7 and tip the balance toward Ang II, which is responsible for vasoconstriction. While the integrity of BBB is tightly regulated by the balance between vasoconstriction and vasodilation, increasing vasoconstriction is unlikely to fully explain the enhanced permeabilization of BBB.  The situation, however, could be explained by the activity of soluble form of ACE2. I suggest that the authors could include the discussion of the post-translational modifications of ACE2 into the manuscript.

The authors cited Ref 53 as a supporting reference for targeting ACE2 as preventive and therapeutic strategies against coronaviruses. Ref 53 refers to use of soluble form of ACE2 to inhibit SARS-CoV-2 infection which has never been successful in treating COVID19 patients. Therefore, the authors should consider citing better alternatives to support the targeting of ACE2.

Some paragraphs, such as Section 6.3, are composed of only 2-3 sentences, making the article appear fragmented. Therefore, the authors should consider combining these paragraphs with others to improve the flow of the article.

Comments on the Quality of English Language

Nil

Author Response

Response to Reviewer’s Comment

First of all, we would like to thank the reviewers for their invaluable suggestions and positive comments on the manuscript. We carefully examined these comments and have revised the manuscript point-by-point along the lines suggested.

The corrections in the revised manuscript are marked using blue markers.

Point 1:

The article focuses on discussing the roles of ACE2 in SARS-CoV-2-induced neurological disorders. The topic is interesting and could be relevant to the development of long COVID. The authors should consider discussing long COVID in the writing.

Response 1: We appreciate the suggestion and have added a new section discussing ACE2 and long COVID. We have made the following changes:

Line 328-344, 6.3 ACE2 and Neurological Symptoms of Long COVID

Multiple studies have reported the persistence of long-term symptoms following COVID-19 infection in some patients, including fatigue, dyspnea, headaches, musculoskeletal pain, altered taste and smell, brain fog, and cognitive decline (such as memory and reasoning difficulties), collectively known as "long COVID" [113-117]. The World Health Organization defines long COVID as symptoms that persist or emerge within three months of the initial SARS-CoV-2 infection, last for at least two months, and cannot be explained by any other cause [118]. Although the precise mechanisms underlying long COVID are not yet fully understood, existing research suggests that the lingering effects of ACE2 following SARS-CoV-2 infection may be closely associated with the neurological symptoms of long COVID [119]. Further studies are needed to explore the long-term alterations in ACE2 following viral infection and how these changes affect the homeostasis and repair processes of the nervous system. Specifically, ACE2's vital roles in maintaining neurological homeostasis, such as regulating inflammatory responses, mitigating oxidative stress, and balancing neurotransmitter levels, may significantly contribute to the pathophysiological processes of long COVID. Additionally, therapies targeting ACE2 could offer promising treatment strategies for patients suffering from long COVID [120].

Point 2:

ACE2 can be proteolytically by proteases, including ADAM17 and TMPRSS2, resulting in shedding the soluble form of ACE2. This form of ACE2 is important for the host to disseminate long-range signaling. Studies have shown that the downregulation of ACE2 expression could be due to the shedding of ACE2 by the activities of these proteases. Furthermore, clinical studies have detected a positive correlation between the detection of the soluble form of ACE2 and the severity of COVID-19. Therefore, the assumption of SARS-CoV-2 infection causes the downmodulation of the RAS signaling pathway could be inaccurate, as the soluble form of ACE2 is functionally active.

Response 2: Thank you for pointing out the importance of the soluble form of ACE2. We have revised Section 5 to discuss the role of proteases like TMPRSS2 in shedding soluble ACE2 and its functional relevance. This revision provides a more accurate representation of the signaling complexity involving ACE2 in the context of COVID-19:

Line 248-254, Furthermore, SARS-CoV-2 infection may cause the shedding of ACE2 into its soluble form (sACE2) via the hydrolytic action of proteases such as TMPRSS2. This process could have significant implications for viral transmission and host signaling [106]. While the exact role of sACE2 remains under investigation, it is likely crucial in regulating the RAS and influencing COVID-19 severity. Therefore, both the membrane-bound and soluble forms of ACE2 should be comprehensively considered when evaluating the impact of SARS-CoV-2 infection on the nervous system.

Point 3:

The authors cited Ref 21 to support the claim that the ACE2 expression could be detected in almost all human organs to various degree is inaccurate. Multiple studies have found that the expression of ACE2 is very restricted. It is recommended that the authors can reference data from general public databases, including https://www.proteinatlas.org/ and https://www.ncbi.nlm.nih.gov/geo/. However, the expression profile may not fully explain the functional sites of ACE2, as the soluble form of ACE2 could be functioning remotely in tissues that originally do not express ACE2. Indeed, some studies have shown that the soluble form of ACE2 could mediate SARS-CoV-2 infection in cells that do not express cellular ACE2. The authors should also discuss this in the manuscript.

Response 3: We have replaced the citation of Ref 21 with data from Protein Atlas, providing a more accurate representation of ACE2’s tissue expression. Additionally, we have clarified that the soluble form of ACE2 may mediate long-range signaling, including in tissues not directly expressing ACE2:

Line 51-53, The expression of ACE2 in healthy human tissues has been described by databases such as Protein Atlas, with the highest expression detected in the intestine, testes, lungs, cornea, heart, kidneys, and adipose tissue [21, 22].

Line 395-404, While the expression profile of ACE2 reveals its distribution across various tissues, the functional role of ACE2 extends beyond tissue-specific expression. Notably, sACE2 can be released into circulation, enabling SARS-CoV-2 infection even in tissues with limited ACE2 expression [121]. Studies have demonstrated that sACE2 can facilitate viral entry into cells lacking cellular ACE2, which has important implications for viral transmission and infection mechanisms [122]. This finding underscores the complexity of ACE2's functional roles and emphasizes the need for further investigation into the mechanisms that enable its distant functionality. Consequently, while the tissue-specific expression of ACE2 is significant, understanding the role of its soluble form in disease pathogenesis is equally essential.

Point 4:

Discussion 6.1. mentioned that “By catalyzing the conversion of Ang II to Ang 1-7, ACE2 balances the vasoconstrictive, pro-inflammatory, and profibrotic effects of Ang II, promoting vasodilation, anti-inflammation, and antifibrosis.” The statement is true about the function of Ang1-7 which is involved in vasodilation, anti-inflammation, and antifibrosis. However, SARS-CoV-2 infection decreases the expression level of ACE2, which should reduce the level of Ang1-7 and tip the balance toward Ang II, which is responsible for vasoconstriction. While the integrity of BBB is tightly regulated by the balance between vasoconstriction and vasodilation, increasing vasoconstriction is unlikely to fully explain the enhanced permeabilization of BBB. The situation, however, could be explained by the activity of soluble form of ACE2. I suggest that the authors could include the discussion of the post-translational modifications of ACE2 into the manuscript.

Response 4: We have expanded the discussion in Section 5 to include details on the role of the soluble form of ACE2 in regulating the Ang II/Ang1-7 balance and its potential impact on the BBB. This revision incorporates the suggestion to discuss post-translational modifications of ACE2 and their relevance to neurovascular regulation:

Line 257-278, When investigating the role of ACE2 in SARS-CoV-2 infection, it is essential to consider not only its function as a viral receptor but also the regulation of its expression and downstream effects in pathological conditions [107]. ACE2 activity is influenced by its expression levels and post-translational modifications, including glycosylation, phosphorylation, and ubiquitination. These modifications can affect ACE2 stability, subcellular localization, and binding affinity to ligands or interacting proteins, indirectly impacting the balance between Ang II and Ang 1-7. SARS-CoV-2 infection may further exacerbate the imbalance between Ang II and Ang 1-7 by disrupting the post-translational modification process of ACE2 [108]. Additionally, sACE2, an enzymatic product of ACE2, plays a crucial role in regulating the Ang II/Ang 1-7 balance. sACE2 can diffuse freely in circulation and competitively bind Ang II with membrane-bound ACE2, modulating the activity of Ang II and its downstream effects [109]. SARS-CoV-2 infection may also alter the balance between Ang II and Ang 1-7 by affecting sACE2 production or function, which in turn influences the permeability of the BBB. The increased permeability of the BBB may involve the coordinated action of several mechanisms, including ACE2 downregulation, elevated Ang II levels, release of inflammatory mediators, and disruption of tight junctions [110-112]. Thus, understanding the impact of SARS-CoV-2 infection on BBB permeability requires comprehensive analysis of ACE2 expression regulation, post-translational modifications, sACE2 activity, and inflammatory mediator interactions. These factors collectively contribute to BBB dysfunction and the neurological complications associated with infection.

Point 5:

The authors cited Ref 53 as a supporting reference for targeting ACE2 as preventive and therapeutic strategies against coronaviruses. Ref 53 refers to use of soluble form of ACE2 to inhibit SARS-CoV-2 infection which has never been successful in treating COVID19 patients. Therefore, the authors should consider citing better alternatives to support the targeting of ACE2.

Response 5: We have replaced the references. The content is as follows:

  1. Jackson CB, Farzan M, Chen B, Choe H. Mechanisms of SARS-CoV-2 entry into cells. Nat Rev Mol Cell Biol 2022; 23: 3-20.
  2. Dang F, Bai L, Dong J et al. USP2 inhibition prevents infection with ACE2-dependent coronaviruses in vitro and is protective against SARS-CoV-2 in mice. Sci Transl Med 2023; 15: eadh7668.

Point 6: Some paragraphs, such as Section 6.3, are composed of only 2-3 sentences, making the article appear fragmented. Therefore, the authors should consider combining these paragraphs with others to improve the flow of the article.

Response 6: We have revised the manuscript. This revision improves the narrative flow and ensures that the discussion is more cohesive and structured.

Reviewer 2 Report

Comments and Suggestions for Authors

The paper is very good and informative. As a review, it offers the reader an exhaustive source of information concerning the ACE 2 and it's functions and puts in perspective In regard to the functionality of the nervous system and the pathogenicity of neurodegenerative diseases. It also showcases the importance of ACE 2 in the evolution of Sars Cov infection.

However, There is some confusion throughout the text between the the ACE2 itself (a protein-enzyme) and it's membrane receptor, which can be present throughout the CNS. It is stated all over in the paper that ACE 2 is the site of the coupling of the virus with the neural or glial or endothelial cells. While the ACE 2 is a free interstitial protein,it is difficult to act as a receptor for a virus . T The so-called ACE2 receptor, it is what ? A membrane protein which binds ACE2 ? Please review and clarify throughout the text if ACE2 is the receptor for the virus.

Author Response

Response to Reviewer’s Comment

First of all, we would like to thank the reviewers for their invaluable suggestions and positive comments on the manuscript. We carefully examined these comments and have revised the manuscript point-by-point along the lines suggested.

The corrections in the revised manuscript are marked using blue markers.

Point 1:

However, There is some confusion throughout the text between the the ACE2 itself (a protein-enzyme) and it's membrane receptor, which can be present throughout the CNS. It is stated all over in the paper that ACE 2 is the site of the coupling of the virus with the neural or glial or endothelial cells. While the ACE 2 is a free interstitial protein, it is difficult to act as a receptor for a virus. The so-called ACE2 receptor, it is what? A membrane protein which binds ACE2? Please review and clarify throughout the text if ACE2 is the receptor for the virus.

Response 1: We appreciate the suggestion, and based on it, I have made corresponding modifications and clarifications to the manuscript. The specific modifications are as follows:

Line 33-36, Angiotensin-converting enzyme 2 (ACE2) is a critical membrane-bound receptor and enzyme that serves dual functions in human physiology and viral infections. As a receptor, ACE2 is essential for the entry of several human coronaviruses.

Line 108-114, ACE2 plays a critical role as a receptor in the infection process of coronaviruses, including SARS-CoV and SARS-CoV-2. It is a membrane-bound protein that is expressed on the surface of cells in various tissues, such as the central nervous system. The S protein on the surface of these viruses binds to ACE2 receptors on host cells, initiating the fusion of viral and host cell membranes, which allows the virus to enter the host cell for replication and spread [11, 53]. Consequently, ACE2 is a significant target for developing strategies to prevent and treat coronavirus infections [54, 55].
